# Cholesterol Regulates Airway Epithelial Cell Differentiation by Inhibiting p53 Nuclear Translocation

**DOI:** 10.3390/ijms26178324

**Published:** 2025-08-27

**Authors:** Ashesh Chakraborty, Juliana Giraldo-Arias, Juliane Merl-Pham, Elisabeth Dick, Michal Mastalerz, Marie Zöller, Hannah Marchi, Ronan Le Gleut, Rudolf A. Hatz, Jürgen Behr, Anne Hilgendorff, Stefanie M. Hauck, Claudia A. Staab-Weijnitz

**Affiliations:** 1Institute of Lung Health and Immunity and Comprehensive Pneumology Center with the CPC-M bioArchive, Ludwig-Maximilians-Universität München and Helmholtz Zentrum München, Member of the German Center for Lung Research (DZL), 85764 Munich, Germanyclaudia.staab-weijnitz@cuanschutz.edu (C.A.S.-W.); 2Metabolomics and Proteomics Core, Helmholtz Zentrum München, 85764 Neuherberg, Germany; 3Core Facility Statistical Consulting, Helmholtz Zentrum München, 85764 Neuherberg, Germany; 4Faculty of Business Administration and Economics, Bielefeld University, 33615 Bielefeld, Germany; 5Thoraxchirurgisches Zentrum, Klinik für Allgemeine-, Viszeral-, Transplantations-, Gefäß- und Thoraxchirurgie, LMU Klinikum, Ludwig-Maximilians-Universität München, 80336 Munich, Germany; 6Department of Medicine V, LMU Klinikum, Ludwig-Maximilians-Universität München, Member of the German Center of Lung Research (DZL), 81377 Munich, Germany; 7University of Colorado, Anschutz Medical Campus, Department of Pediatrics and Division of Pulmonary Sciences and Critical Care Medicine, School of Medicine, Aurora, CO 80045, USA

**Keywords:** cholesterol, primary human bronchial epithelial cells, p53, Ingenuity Pathway Analysis

## Abstract

Cholesterol is an essential plasma membrane component, and altered cholesterol metabolism has been linked to cholesterol accumulation in the airways of COPD and cystic fibrosis patients. However, its role in airway epithelial differentiation is not well understood. Tandem mass spectrometry-based proteomic analysis of differentiating primary human bronchial epithelial cells (phBECs) revealed an overall inhibition of the cholesterol biosynthesis pathway. We hypothesized that excess cholesterol impairs the differentiation of phBECs into a fully functional bronchial epithelium. PhBECs were differentiated in the presence of 80 µM cholesterol for 21 days, the main airway cell type populations monitored using qRT-PCR and immunofluorescent stainings, and epithelial barrier integrity was analyzed via transepithelial electrical resistance measurements. Chronic cholesterol exposure led to a significant increase in CC10^+^ secretory cells at the expense of ciliated cells. Pathway enrichment analysis suggested the tumor protein p53 as a master regulator of genes during normal differentiation of phBECs. Chronic cholesterol exposure drastically impaired the nuclear translocation of p53. Our findings suggest that this inhibition underlies the cholesterol-induced expansion of CC10^+^ secretory cell populations at the expense of ciliated cells. In conclusion, we identify cholesterol as an important regulator of normal bronchial epithelial cell differentiation through inhibition of p53 nuclear translocation.

## 1. Introduction

In human conducting airways, progenitor basal cells give rise to a fully differentiated pseudostratified epithelium consisting of ciliated cells, secretory cells, including goblet cells and club cells, and basal cells. Secretory cells are non-ciliated differentiated cell types present within the pseudostratified bronchial epithelium and typically exhibit a cuboidal to columnar shape. Ciliated cells are generally columnar in shape, extending to the luminal surface, while basal cells are located lining the basement membrane of the human airway epithelium. Furthermore, the human airway epithelium also contains some rare cell types, such as tuft cells, ionocytes, and neuroendocrine cells, which collectively remain below 1% [1,2]. The airway epithelium forms a protective barrier and facilitates mucociliary clearance to defend against inhaled toxic insults [1,2,3,4]. This differentiation process can be fully recapitulated in vitro, where progenitor basal stem cells give rise to a fully functional bronchial epithelium containing all major cell types stated above [3,4]. Furthermore, molecular signaling pathways, such as transforming growth factor beta (TGF-β), bone morphogenic protein (BMP), WNT, sonic hedgehog (SHH), and Notch signaling pathways, play an important role in the maintenance of progenitor basal stem cells in the airways. These signaling pathways modulate proliferation and differentiation during the developmental stages of the lung and remain largely quiescent postnatally [2,5]. However, they undergo reactivation after injury and initiate airway repair by inducing proliferation and differentiation of the progenitor basal stem cells in the airways.

Aberrant epithelial repair driven by the reactivation of developmental signaling pathways and the activation of progenitor basal stem cells has been implicated in both acute and chronic lung injury [2,6]. However, the underlying mechanisms governing adult human airway regeneration remain poorly understood. In this context, elucidating the normal differentiation processes of human progenitor basal stem cells into fully functional airway epithelium serves as a model for airway regeneration. While previous studies have highlighted transcriptome changes during the differentiation of basal cells into the airway epithelium [7,8], by far, not all transcript changes are translated to the protein level, which better represents cell phenotype and function [9].

In the human conducting airways, cholesterol (CHOL) is transported through the bloodstream primarily bound to carrier proteins, such as low-density lipoprotein (LDL) and high-density lipoprotein (HDL), which then interact with bronchial epithelial cells via specific receptor-mediated mechanisms to facilitate uptake and subsequent intracellular synthesis [10,11,12,13]. However, the role of human bronchial epithelial cell subtypes in synthesizing CHOL remains unknown to date. In the human alveolar compartment, CHOL constitutes a major neutral lipid component of the pulmonary surfactant, modulating surfactant membrane structure and maintaining reduced surface tension of small airways and alveoli [14,15,16]. However, impaired CHOL metabolism and CHOL overload have been implicated in several lung diseases. In chronic obstructive pulmonary disease (COPD), increased lipid-laden alveolar macrophages have been observed in bronchoalveolar lavage fluids from COPD patients [17,18,19,20] and hypercholesterolemia in COPD patients [11]. Jia et al. reported that oxysterol, a metabolite of CHOL, is involved in the formation of inducible bronchus-associated lymphoid tissue (iBALT), a key contributor to COPD pathogenesis observed in COPD patients [21]. In cystic fibrosis, mutations in the cystic fibrosis transmembrane conductance regulator (CFTR) alter lipid metabolism and are associated with excess CHOL accumulation [22,23]. Furthermore, in pulmonary alveolar proteinosis, a rare lung disease caused by increased accumulation of pulmonary surfactant in the alveolar region, an elevated CHOL content has been observed in the accumulated surfactant [24]. Moreover, an increasing body of evidence suggests that CHOL uptake at the cell membrane plays a pivotal role in facilitating viral entry and assembly, enhancing the infection of viruses, including SARS-CoV-2 [25,26] and human rhinovirus [27]. In summary, excessive CHOL accumulation resulting from lipid dysregulation is a common feature of acute and chronic lung diseases, but its impact on human airway basal cell differentiation remains unexplored.

Here, we conducted a comprehensive liquid chromatography-tandem mass spectrometry (LC-MS/MS)-based proteomic analysis across the full-time course of primary human bronchial epithelial cells (phBECs) differentiation to identify novel key regulators of normal airway differentiation using pathway enrichment analysis. This analysis highlighted CHOL and p53 as potential regulators and prompted us to investigate their roles in our culture system. We show that chronic CHOL exposure led to a significant rise in the club cell-specific protein 10 (CC10^+^) secretory cell population during the differentiation phase. Pathway enrichment analysis of differentially expressed proteins from normal differentiating phBECs predicted tumor protein 53 (p53) as a master regulator during normal healthy phBECs differentiation, including inhibition of CHOL biosynthesis. As loss of p53 has previously been associated with an increase in CC10^+^ secretory cells and a reduction in ciliated cells [28], our finding that chronic CHOL exposure inhibited the nuclear translocation of p53 supports the idea that p53 inactivation may drive the observed shifts in cell type populations.

Some of the results of this study have been previously reported in the form of a conference abstract [29].

## 2. Results

### 2.1. MS/MS-Based Proteomic Analysis of Normal Differentiating phBECs Reveals Inhibition of the CHOL Biosynthesis Pathway

Using label-free quantitative LC-MS/MS-based analysis, we profiled proteome changes at five distinct timepoints (day 0, 7, 14, 21, and 28) throughout the full course of normal phBEC differentiation (Figure 1A).

At day 28 of the differentiation phase, the proteomic analysis identified a total of 1650 differentially expressed proteins out of 4860 detected proteins, relative to the baseline at day 0. The list of the top 100 differentially regulated proteins during the entire phBECs differentiation phase is shown in Appendix A. Next, to gain insight into the biological pathways modulated during the differentiation process, the differentially expressed proteins were subjected to Ingenuity Pathway Analysis (IPA version 2023.1) software for pathway enrichment analysis. This analysis revealed a total of 534 deregulated signaling pathways, including key developmental signaling pathways, such as TGF-β, BMP, WNT, and SHH, as well as pathways associated with cell death and detoxification mechanisms. Interestingly, the top five downregulated signaling pathways among those, zymosterol biosynthesis, superpathway of CHOL biosynthesis, and CHOL biosynthesis pathway I, II, and III, all indicated a consistent loss of CHOL biosynthesis during normal phBEC differentiation (Figure 1B). More specifically, phBEC differentiation was accompanied by persistent downregulation of key proteins involved in the CHOL biosynthesis pathway during the entire differentiation phase: squalene epoxidase (SQLE), a key rate-limiting enzyme in CHOL biosynthesis that converts squalene to 2,3-oxidosqualene, showed sustained downregulation during the differentiation phase. Also, other critical enzymes in the CHOL biosynthesis pathway, including farnesyl diphosphate synthase (FDPS), mevalonate diphosphate decarboxylase (MVD), 3-hydroxy-3-methylglutaryl-CoA synthase 1 (HMGCS1), and farnesyl-diphosphate farnesyltransferase 1 (FDFT1, also known as squalene synthase), were consistently downregulated during the differentiation phase (Figure 1C,D). All CHOL-metabolizing proteins were identified with at least two matching peptides, including one unique peptide, and Mascot confidence scores of ≥47 (Table 1), reflecting the high specificity and reproducibility of this approach, which typically exceeds that of antibody-based methods. Based on these findings, our proteomic analysis suggested an overall inhibition of the CHOL biosynthesis pathway as a key event governing bronchial epithelial cell fate during the differentiation phase.

Having identified CHOL as a potential key regulator of bronchial epithelial cell fate, we sought to investigate how chronic exposure to CHOL influences airway epithelial cell differentiation in vitro. We used CHOL to treat phBECs chronically via the basolateral compartment of the insert at a concentration of 80 µM. This mode of chronic CHOL exposure via the basolateral compartment mimics the human physiological route, where CHOL bound to carrier proteins, such as LDL and HDL, circulates in the bloodstream and is subsequently taken up by airway epithelial cells through specific receptors [10,20]. Moreover, it allows for prolonged CHOL exposure without compromising the ALI condition [30]. During chronic CHOL exposure, the airway epithelial barrier integrity was determined using transepithelial electrical resistance (TEER). In addition, selected cell type-specific markers were determined at the transcript level using qRT-PCR and at the protein level using immunofluorescence (IF) stainings for the indicated time points during the differentiation phase (Figure 2A). Throughout the entire differentiation phase, chronic exposure with 80 µM CHOL resulted in cytotoxicity below 20%, with no significant difference compared to the vehicle control (Appendix A).

Upon chronic CHOL exposure, the TEER analysis exhibited intact barrier integrity with values ranging between 400 and 800 Ω cm^2^. We did not observe any impact of chronic CHOL exposure on the epithelial barrier integrity in comparison to time-matched vehicle control until day 14, but we observed a slight decrease (*p* value = 0.12) in TEER values in the chronic CHOL-treated phBECs as compared to the vehicle control at day 21 of the differentiation phase (Figure 2B). Next, we determined the effect of chronic CHOL exposure on the cell type-specific transcripts using qRT-PCR analysis. For this, after chronic CHOL exposure, we determined transcript levels using well-established human bronchial epithelial cell type-specific markers, such as the ciliated cell marker *FOXJ1*, the secretory cell markers *MUC5AC* and *SCGB1A1*, basal cell markers *TP63* and *KRT5* [3,4,30], and the squamous cell marker *IVL*, and observed no specific changes in these transcripts after chronic CHOL exposure in comparison to time-matched vehicle control during the differentiation phase (Figure 2C).

Additionally, after chronic CHOL exposure, we also determined the bronchial epithelial cell type composition by immunofluorescent staining of cell type-specific markers, such as alpha-tubulin (αTUB, ciliated cell), mucin 5AC (MUC5AC), club cell-specific protein 10 (CC10, encoded by *SCGB1A1*), and tumor protein 63 (p63, encoded by *TP63*), for the indicated time points during the differentiation phase (Figure 3A). During the differentiation phase, we observed an increase in absolute total cell count in a time-dependent manner, indicating proliferation. Alongside the rise in cell number, we also observed increased absolute cell counts for MUC5AC^+^ and CC10^+^ secretory cells, αTUB^+^ ciliated cells, and p63^+^ basal cells in both vehicle control and chronic CHOL exposure conditions. As a result of chronic CHOL exposure, we observed an increase in CC10^+^ secretory cell counts relative to the time-matched vehicle control during the entire differentiation phase, which reached statistical significance for time points day 7 and day 21 (Figure 3B). As for other differentiated cell types, the absolute cell counts for p63^+^ basal cells and MUC5AC^+^ secretory cells remained unchanged in both exposure conditions during the differentiation phase. At day 21 of the differentiation phase, a slight but not statistically significant decrease (*p* value = 0.26) in the absolute cell count of the αTUB^+^ ciliated cell population was observed in the chronic CHOL exposure condition as compared to the time-matched vehicle control (Figure 3A,B). Furthermore, the numerical decrease in αTUB^+^ ciliated cell count observed following chronic CHOL exposure corresponds to the concomitant increase in the CC10^+^ secretory cell count (Figure 3B). Overall, this suggests that chronic CHOL exposure during the differentiation phase increased the CC10^+^ secretory cell population at the expense of ciliated cells without affecting airway epithelial barrier integrity.

### 2.2. Chronic CHOL Exposure Impairs Nuclear Translocation of p53 In Vitro

Next to the list of activated and deactivated pathways described above, our pathway enrichment analysis also predicted a list of potential upstream transcription regulators, such as tumor protein 53 (p53), interleukin enhancer-binding factor 3 (ILF3), sterol regulatory element-binding protein 1 (SREBF1), and nuclear protein 1 (NUPR1) (Table 2). Among these, the transcription regulator p53 was predicted as the top master regulator for bronchial epithelial cell differentiation (Table 2).

Given that loss of p53 has been described to increase the self-renewal of CC10^+^ cells and decrease the ciliated cell population [28], we hypothesized that CHOL impairs p53 translocation to the nucleus. We therefore assessed its subcellular localization in differentiating phBECs using immunofluorescence analysis (Figure 4A). In control cells, the p53 protein predominantly localized to the nucleus, indicating active p53 signaling throughout the differentiation phase. In contrast, chronic CHOL exposure resulted in a drastic reduction in nuclear p53 localization at days 14 and 21, suggesting impaired p53 signaling in comparison to the time-matched vehicle control (Figure 4B). Therefore, chronic CHOL exposure alters the subcellular localization of p53 in differentiating phBECs, promoting a shift from nuclear to cytoplasmic localization during differentiation. This indicates that CHOL exposure impairs p53-dependent gene transcription and subsequent cellular processes and supports the idea that this inhibition underlies the observed changes in CC10^+^ secretory and ciliated cell populations.

## 3. Discussion

Proteomic profiling of differentiating phBECs in vitro followed by pathway enrichment analysis revealed a global repression of the CHOL biosynthesis pathway during the differentiation process and predicted p53 activation as a key regulator of bronchial epithelial differentiation. Supplementation with exogenous CHOL during differentiation markedly inhibited nuclear translocation of p53 and resulted in an increase in CC10^+^ secretory cells and a trend of concomitant decrease in αTUB^+^ ciliated cells. These findings suggest that chronic CHOL exposure impairs p53-dependent regulation of human bronchial epithelial cell fate.

Pathway enrichment analysis suggested p53 to be the top master regulator of normal airway epithelial cell differentiation in our in vitro system (Table 2). While p53 is well-known for its function as a tumor suppressor, it has only more recently been described as an important factor in the maintenance of airway epithelial cell density and cell type composition. p53 plays an important role in maintaining the balance between secretory and ciliated cell populations, while its loss is associated with increased progenitor and self-renewal capacity of CC10^+^ secretory cells and a reduced ciliated cell population [28]. In our in vitro system, chronic CHOL significantly impaired the nuclear translocation of p53, suggesting that this p53 inhibition at the same time led to increased progenitor CC10^+^ secretory cells and a trend of reduced αTUB^+^ ciliated cells (Figure 3 and Figure 4). Notably, inhibition of the Notch signaling pathway is well-established to increase secretory and reduce ciliated cell numbers in human and mouse airway epithelium [30,31,32,33,34]. Interestingly, elevated CHOL levels have been shown to inhibit the Notch signaling pathway [35,36], raising the possibility that CHOL inhibits both p53 and Notch signaling pathways. Given that the inhibition of either pathway results in increased CC10^+^ secretory club cells and reduced αTUB+ ciliated cells, our results warrant the exploration of the crosstalk between p53 and Notch in mediating CHOL-related effects on cell type composition and stem cell potential [37]. With p53 implicated in cell cycle growth arrest, senescence, apoptosis, and tumor suppression [38] and CC10^+^ secretory club cells significantly enriched in aging and COPD pathogenesis [39], our findings warrant future research into the molecular pathways through which p53 inhibition may lead to an increased CC10^+^ secretory cell population in the context of cellular senescence, aging, and carcinogenesis.

At first, intending to treat phBECs with CHOL levels found in the human blood plasma, we wanted to use two different CHOL concentrations defined as physiological normal and pathological levels corresponding to 100 mg/dl (2.58 mM) and 500 mg/dl (12.9 mM), respectively [40]. However, in this study, due to the poor solubility of CHOL and the toxicity of ethanol-based solvents in human airway epithelial cells [30], the maximal achievable concentration of CHOL for prolonged basolateral exposure in our in vitro system was limited to 80 µM. As CHOL is predominantly transported in the bloodstream bound to carrier proteins, a direct comparison to physiological free CHOL concentrations is challenging, but free serum CHOL concentrations in the range from 1.4 to 1.6 mM have been reported [41]. Therefore, the concentration used in this study can be considered at the lower end of the physiological range. Interestingly, phBECs were highly susceptible to such a comparatively low concentration of CHOL, as it caused not only a significant induction in CC10^+^ secretory cells at the expense of αTUB^+^ ciliated cells (Figure 3) but also inhibition of p53 nuclear translocation in airway epithelial cells. These findings suggest that the downregulation of the CHOL biosynthesis pathway is a critical mechanism for maintaining airway epithelial homeostasis and preserving proper cell type composition within the human airway epithelium. So far, several studies have reported increased CHOL levels in chronic lung diseased patients’ lungs [17,18,19,20,21,22,23,24,25,26,27], but only a single study has reported direct effects of CHOL on airway epithelium, namely that accumulation of CHOL sulfate in rabbit tracheal epithelial cells for up to 16 days induced squamous epithelial cell differentiation [42]. Squamous cells are typically thin and flat shaped, lining the basement membrane of the airway epithelium. However, in this study, cells were cultured under submerged culture conditions and not at the ALI, which does not allow for the normal differentiation route, which mimics airway differentiation and regeneration in vivo. Submerged exposure also poorly models systemic exposure of airway epithelial cells, which is better mimicked by basolateral exposure in transwells in our culture system. In our in vitro ALI system, chronic CHOL exposure did not induce a well-established marker of terminal squamous cell differentiation, involucrin (*IVL*) [43,44], during the entire differentiation phase (Figure 2C). We believe that this is due to the more physiologically relevant culture conditions in our study but cannot exclude that the CHOL concentrations used here were insufficient to trigger squamous differentiation.

Interestingly, CHOL depletion has been demonstrated to result in suppression of mucus hypersecretion in COPD patients. Also, statins have been reported to deplete CHOL in the plasma membrane of human airway epithelial cells, such as NCI-H292 cells and Calu-3 cells, and prevent mucus hypersecretion in vitro [45,46]. However, the underlying mechanism as to how CHOL overload in the airway epithelium may contribute to activating airway goblet cells and lead to mucus hypersecretion is unknown. These findings suggest a potential role of CHOL in the regulation of mucin production and indicate that CHOL homeostasis may be a critical determinant of goblet cell activity. In our in vitro study, chronic CHOL exposure did not induce changes in the MUC5AC^+^ goblet cell population (Figure 2C and Figure 3). This suggests that the effects of CHOL on airway epithelial cell differentiation are context-dependent and may, for instance, vary under inflammatory conditions. Further research is required to elucidate the impact of CHOL accumulation on airway goblet cell differentiation and function.

Lipid rafts are CHOL and sphingolipid-enriched microdomains located within the plasma membrane that play important roles in receptor-mediated signaling and nuclear import mechanisms [47]. We speculate that prolonged exposure of phBECs to exogenous CHOL may alter lipid raft composition due to an increase in membrane CHOL content, potentially disrupting the organization and function of key signaling receptors. Notably, there is evidence that the Epidermal Growth Factor Receptor (EGFR) located within lipid rafts regulates cellular growth, proliferation, and differentiation, and functions upstream of the p53 signaling pathway [48,49,50]. Therefore, it would be of great interest to further explore the impact of CHOL-induced lipid raft alterations on EGFR activity and its downstream effects on p53 nuclear import mechanisms.

In the distal lung, the presence of CHOL is crucial for surfactant production, while an increase in the CHOL levels in the proximal lung region has been associated with chronic lung diseases. This suggests that there could be a CHOL concentration gradient that increases towards the distal lung, reaching alveoli, ultimately leading to the relative increase in CC10^+^ secretory cells. Our established in vitro airway epithelium model, which exhibits increased club cells with a trend of concomitant loss of ciliated cells due to chronic CHOL exposure, offers a valuable system for evaluating the effects of CHOL on airway epithelial cell differentiation. This is relevant due to an increase in CHOL levels observed in numerous chronic lung disease conditions [17,18,19,20,21,22,23,24,25,26,27] but also because the altered cell type composition observed may result in impaired mucociliary clearance and defective detox mechanisms against inhaled toxicants, thus increasing susceptibility to lung injury and disease [3,51,52,53].

Notably, it has been shown that the presence of immune cells, such as alveolar macrophages, is involved in regulating lipid metabolism, including CHOL, and maintaining homeostasis [54]. The absence of such immune cells in our in vitro system is a limitation of our study. Therefore, a co-culture in vitro model comprising airway epithelial cells and immune cells, such as macrophages, will provide a more physiologically relevant environment to investigate how immune-epithelial interactions regulate CHOL metabolism.

## 4. Materials and Methods

### 4.1. Reagents and Chemicals

Cholesterol was purchased from Alfa Aesar (57-88-5, St. Louis, MO, USA) and was dissolved in absolute ethanol (64-17-5, Sigma, St. Louis, MO, USA) at a stock concentration of 16 mM.

### 4.2. Patient Material

phBECs from five patients were obtained from the CPC-M bioarchive at the Comprehensive Pneumology Center (CPC, Munich, Germany). phBECs were obtained from histologically normal regions adjacent to resected lung tumors, as described previously [3,30]. The study was approved by the local ethics committee of the Ludwig-Maximilians University of Munich, Germany (Ethics votes #333-10 and #19-630). Written informed consent was obtained from all study participants.

### 4.3. Primary Human Bronchial Epithelial Cell (phBEC) Differentiation

At first, the phBECs were expanded on a type 1 collagen coated at a seeding density of 100,000–150,000 cells on 100 mm plates (Corning, 430167, New York, NY, USA) using PneumaCult-Ex plus medium with 1 × supplement (Stemcell Technologies, 05041, Vancouver, BC, Canada), 0.2% hydrocortisone (Stock: 96 µg/mL) (CAS Number: H2270, Sigma) and 1% Pen/Strep (Life technologies, 10,000 U, 15140, New York, NY, USA). Upon reaching 80–90% confluency, the cells were trypsinized and then seeded on 12-well transwell (Corning: 353180, 10.5 mm inserts; Corning: 353503, 12-well plate, 0.4 µm polyester membrane, tissue culture treated, polystyrene, 0.9 cm^2^/transwell, Durham, NC, USA) at a seeding density of 80,000 cells/insert and were cultured under submerged conditions until they reached confluency. Upon reaching 80–90% confluency under submerged conditions, phBECs were then airlifted by aspirating the apical medium, and the basolateral medium was replaced with Pneumacult-ALI (Stemcell Technologies, 05002, including supplement 05003, and additives 05006, Vancouver, BC, Canada) + 0.2% heparin (stock: 2 mg/mL) (H3149, Sigma) + 0.5% hydrocortisone (stock: 96 µg/mL) + 1% Pen/Strep and was differentiated for 28 days at ALI condition.

### 4.4. Proteome Analysis from Normal Differentiating phBECs

During normal phBECs differentiation at the ALI condition, the inserts containing cells from four independent biological replicates were collected at five different time points (days: 0, 7, 14, 21, and 28). Cells were lysed in ice-chilled DOC lysis buffer (10 mM Tris-Cl pH 7.5, 1% sodiumdeoxycholate (SDC), 1 mM EDTA-Na, +1× complete) using a Precellys homogenizer (Bertin Technologies). Insoluble proteins were collected by centrifugation, solubilized in 6 M guanidinium hydrochloride, 100 mM Tris/HCl, pH 8.5, and reduced, alkylated, precipitated, and digested as described before [55]. Soluble proteins in the supernatant fraction were digested using a modified FASP procedure [56,57].

LC-MSMS measurements were performed on an Ultimate 3000 RSLC (Thermo) online coupled to a Q-Exactive HF or HF-X (Thermo). Approximately 0.5 μg of soluble or insoluble protein sample was loaded onto the trap column (300 µm inner diameter (ID) × 5 mm, packed with Acclaim PepMap100 C18, 5 µm, 100 Å from LC Packings) and after 5 min, eluted and separated on the C18 analytical column (nano-ease M-class HSST3 column, 1.8 μm particles, 25 cm length, Waters) by a 90 min non-linear acetonitrile gradient. MS spectra were recorded at a resolution of 60,000 with an automatic gain control (AGC) target of 3 × 10^6^ and a maximum injection time of 50 or 30 ms from 300 to 1500 *m*/*z*. From the MS scan, the 10 or 15 most abundant peptide ions were selected for fragmentation via higher-energy collisional dissociation with a normalized collision energy of 27, an isolation window of 1.6 *m*/*z*, and a dynamic exclusion of 30 s. MS/MS spectra were recorded at a resolution of 15,000 with an AGC target of 10^5^ and a maximum injection time of 50 ms. Intensity threshold was set to 1 × 10^4^ and unassigned charges and ions with charges of +1 and N + 8 were excluded.

The spectra obtained from LC-MS/MS analysis were uploaded to the Progenesis QI proteomics software (v4.2, Nonlinear Dynamics, part of Waters). The Mascot search engine was used to identify peptides (version 2.6.1) from the Swiss-Prot human protein database, resulting from the MSMS spectra (Release 2017_02, 11,451,954 residues, 20,237 sequences), as described [58,59]. The search parameters were defined as follows: a fragment mass tolerance of 20 mmu, a peptide mass tolerance of 10 ppm, with cysteine carbamidomethylation specified as a fixed modification. Variable modifications allowed for methionine oxidation and glutamine deamidation, and allowed one missed cleavage site. For the insoluble ECM-containing fraction, additionally, lysine and proline oxidation were allowed as variable modifications. A Mascot-integrated decoy database search calculated a false discovery rate of 0.55% or 0.38% when the searches were performed using Mascot percolator score and a significance threshold of 0.05. Peptide assignments were reimported into the Progenesis QI software, and the abundances of all unique peptides allocated to each protein were summed up per soluble and insoluble dataset. Both datasets were then combined to generate a combined protein list with normalized protein abundances per original sample. The mass spectrometry proteomics data have been deposited to the ProteomeXchange Consortium via the PRIDE [1] partner repository with the dataset identifier PXD063763.

### 4.5. Bioinformatic and Ingenuity Pathway Analysis

The differential proteome data from normal differentiating phBECs were performed for four time points (day 7, 14, 21, and 28) as compared to day 0, and the statistical analysis was performed with the R program (Version: R version 4.0.0) using the Wald test with Storey correction to account for multiple testing to identify differentially expressed proteins (significance value q < 0.05). The differentially expressed proteins from normal differentiating phBECs were uploaded to the Ingenuity Pathway Analysis platform (IPA Tool; Ingenuity^®®^Systems, Redwood City, CA, USA; http://www.ingenuity.com) to perform pathway enrichment analysis and identify the upstream master regulator. The parameters were used in IPA to analyze differentially expressed proteins from normal differentiating phBECs with the following criteria: a log fold change threshold of <−1 or >+1, a q-value < 0.05, and statistical significance was determined using Fisher’s exact test *p*-value.

### 4.6. Chronic CHOL Exposure

The CHOL exposure was performed chronically in the basal compartment of the insert for 21 days during normal phBECs differentiation. The 80 µM chronic CHOL concentration used for the exposure was obtained after diluting an absolute ethanol-based CHOL stock in the PneumaCult-ALI medium. The vehicle control used in this study was obtained after adding the same volume of absolute ethanol to the PneumaCult-ALI medium, resulting in a total absolute ethanol concentration of 0.5%.

### 4.7. Lactate Dehydrogenase (LDH) Assay

After chronic CHOL exposure, both apical wash and basolateral medium were collected from the vehicle control and CHOL exposure at days 7, 14, and 21, and stored in 1.5 mL Eppendorf aliquots at −80 °C for further use. The LDH release in the apical and basal supernatant was determined using the cytotoxicity detection kit LDH (Roche, 11644793001, Mannheim, Germany) and used according to the manufacturer’s instructions. For the positive control, the cells were lysed with 2% Triton X/1× HBSS (+Ca^2+^/Mg^2+^) (10× HBSS: 14065049, Gibco, New York, NY, USA) on the apical compartment of the insert for 60 min, and then the apical wash collected was used for maximal LDH release.

### 4.8. RNA Isolation and qRT-PCR Analysis

Upon chronic CHOL exposure, the cells were washed twice with 1× Hank’s Balanced Salt Solution (HBSS) (+Ca^2+^/Mg^2+^) on both the apical and basolateral compartments of the inserts, followed by scraping cells from the insert and collecting the cell lysates into 1.5 mL Eppendorf. The RNA extraction from the collected cell lysates was carried out using the RNeasy Mini Plus Kit (Qiagen, 74136, Venlo, The Netherlands) and used according to the manufacturer’s instructions. The RNA concentration was determined using a NanoDrop 1000 spectrophotometer by measuring absorbance at 260 nm (NanoDrop Tech. Inc; Wilmington, Germany). RNA was reverse transcribed into cDNA using reverse transcriptase (Applied Biosystems, N8080018, Waltham, MA, USA, or Invitrogen, 28025013) and random hexamer primers (Applied Biosystems). A total of 1 µg RNA was diluted up to 20 µL with DNase/RNase-free water, denatured at 70 °C for 10 min for the removal of secondary and tertiary RNA structures, and then later incubated on ice for 5 min. A total of 20 µL of cDNA synthesis master mix (5 mM MgCl_2_, 1× PCR buffer II (10×), 1 mM dGTP, 1 mM dATP, 1 mM dTTP, 1 mM dCTP, 1 U/µL RNase inhibitor, and 2.5 U/µL MuLV reverse transcriptase) was added to each sample and cDNA synthesis was performed for 60 min at 37 °C, followed by a 10 min incubation at 75 °C. The cDNA obtained was then diluted up to 200 µL with DNase/RNase-free water for use during qRT-PCR analysis. qRT-PCR analysis was performed in a 96-well format using a Light Cycler LC480II instrument (Roche) and LightCycler^®®^ 480 DNA SYBR Green I Master (Roche). The data were shown as −ΔCt, where ΔCt was calculated as Ct (gene of interest) − Ct (reference gene) for each condition. WD Repeat Domain 89 (*WDR89*) and a second independent housekeeping gene, hypoxanthine guanine phosphoribosyl transferase (*HPRT*), were used as a housekeeping gene for samples obtained from CHOL-treated phBECs.

### 4.9. Immunofluorescence (IF) Analysis and Quantification

After chronic CHOL exposure, the inserts were washed twice on both apical and basal sides with 1× HBSS (+Ca^2+^/Mg^2+^), followed by cell fixation using 3.7% paraformaldehyde (PFA) for 1 h at room temperature. After fixation, the inserts were washed with 1× DPBS (10× DPBS: 14080-055, Gibco) and stored in 1× DPBS at 4 °C for further use. The membrane-containing cells were cut into six pieces and were first permeabilized with 0.2% Triton X-100/1× DPBS for 5 min, followed by washing with 1× DPBS for 5 min and blocked with 5% BSA/1× DPBS for 1 h at room temperature. After blocking, the membrane-containing cells were transferred to a 12-well plate, and the primary antibody was applied in a total volume of 150 µL and left at 4 °C overnight. The next day, the membranes were washed three times with 1× DPBS for 5 min, and the secondary antibodies conjugated with AlexaFluor-488 and -568 together with 0.5 µg/mL 4,6-diamidino-2-phenylindole (DAPI) (1:2000 dilution) in 5% BSA/1× DPBS were added and incubated for 1 h at room temperature covered in the dark. After secondary antibody incubation, the membrane-containing cells were washed three times with 1× DPBS and mounted in a fluorescent mounting medium (Dako, S3023, Hamburg, Germany).

The immunofluorescence analysis was carried out using a fluorescent microscope (Axiovert II; Carl Zeiss AG; Oberkochen, Germany). At least three representative images from each condition and cell line were obtained using a 20× objective (dimensions: 447.64 µm × 335.40 µm), and the quantification of these images was performed using Imaris 7.4.0 software 6 (Bitplane). In the Imaris quantification software, the spot count function was used to quantify cell populations and was presented as absolute cell numbers.

### 4.10. Primers and Antibodies

The list of primers used in this study was obtained from Eurofins Genomics Germany GmbH (Ebersberg, Germany) and is listed in Table 3. The antibodies used in this study are listed in Table 4.

### 4.11. Trans Epithelial Electrical Resistance (TEER) Measurement

The airway epithelial barrier integrity of phBECs chronically treated with CHOL was assessed at days 7, 14, and 21. At first, the pre-warmed 1× HBSS (+Ca^2+^/Mg^2+^) was added onto the apical compartment of the insert and equilibrated at room temperature for 10 min. After equilibration, the measurement was performed in triplicate for each insert using a Millicell-ERS-2 volt-ohmmeter (Millipore, Billerica, MA, USA) with an STX01 chopstick electrode (Millipore). For the final TEER measurement value, the blank value (measurement of an insert without cells) was subtracted from the determined TEER value, and the resulting value was then multiplied by the surface area of the insert (0.9 cm^2^ for 12-well transwell inserts), and the final value was presented in Ω × cm^2^.

### 4.12. Statistical Analysis

In this study, the results are obtained from five independent biological replicates unless mentioned otherwise, and the data presented here are in terms of mean ± SEM. The statistical analysis for all data was performed using a two-tailed paired *t*-test with Bonferroni correction comparing chronic CHOL-treated to time-matched vehicle control-treated phBECs during the differentiation phase, using GraphPad Prism version 10.2.3 for Windows, GraphPad Software, San Diego, CA, USA, www.graphpad.com.

## 5. Conclusions

Chronic CHOL exposure strongly inhibited nuclear translocation of p53 and resulted in a significant increase in the CC10^+^ secretory cell population at the expense of ciliated cells. Our study thus identifies CHOL as an important regulator of normal epithelial bronchial differentiation and suggests that CHOL may exert these effects via inhibition of p53 nuclear translocation.

## Figures and Tables

**Figure 1 ijms-26-08324-f001:**
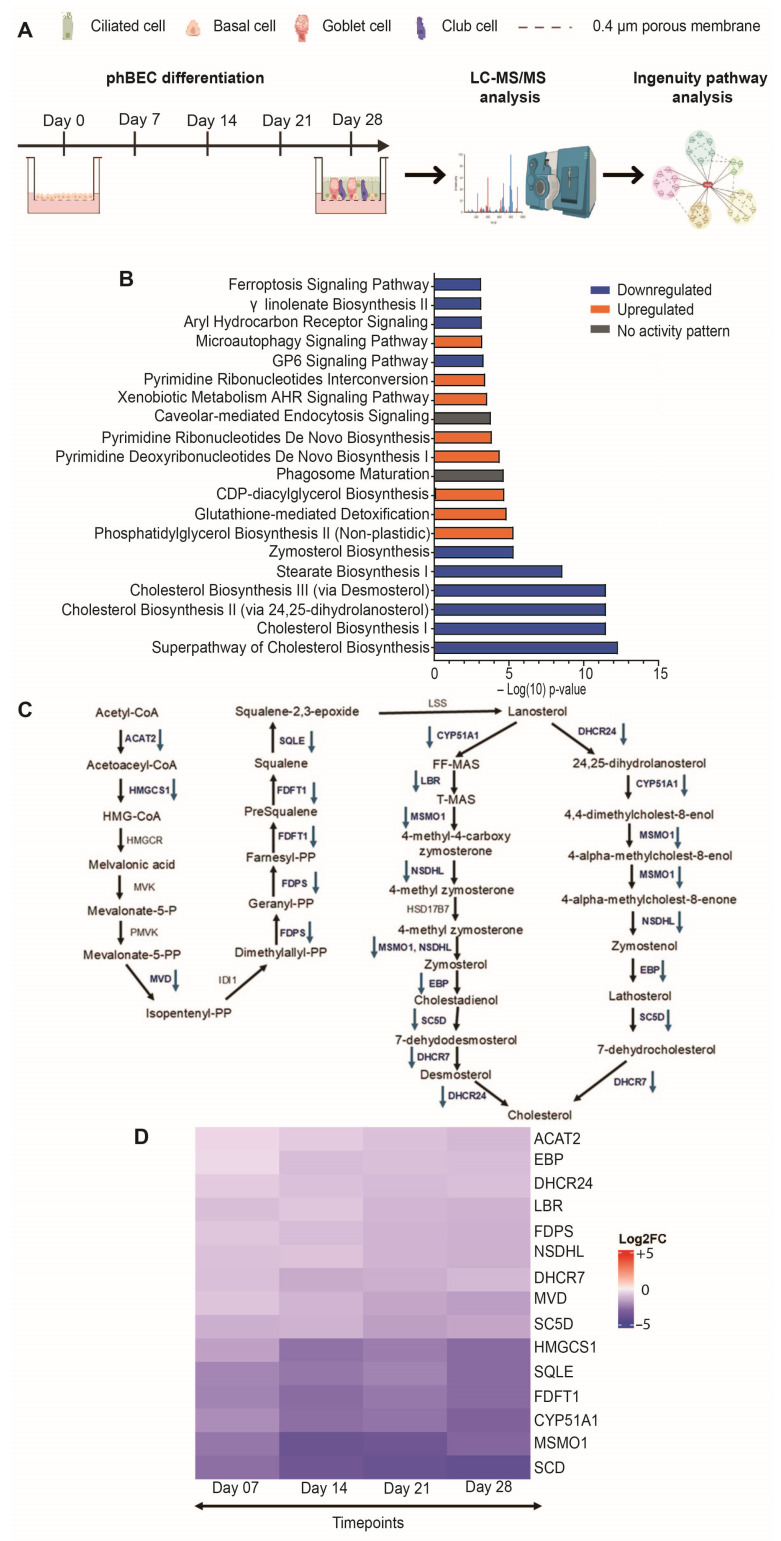
Inhibition of CHOL biosynthesis pathway in normal differentiating phBECs in vitro. (**A**) Schematic overview of the proteomic analysis of normal phBECs differentiated at the air-liquid interface (ALI) condition for 28 days. Figure created using Biorender. (**B**) Top 20 deregulated signaling pathways from differentially expressed proteins. “Upregulated” or “Downregulated” indicates that the proteins within the pathway are either upregulated or downregulated, respectively, suggesting a consistent alteration in pathway activity, whereas “No activity pattern” indicates that proteins within the pathway are altered, but not in a consistent way that would suggest overall up- or downregulation of the pathway. (**C**) Detailed overview of the cholesterol biosynthesis pathway. Proteins involved in the CHOL biosynthesis pathway that were differentially expressed and downregulated during normal phBECs differentiation are indicated with downward arrows. Undetected proteins within the pathway are shown without any specific marking. (**D**) List of differentially expressed protein profiles involved in the CHOL biosynthesis pathway during the entire differentiation phase. Data shown are derived from five independent biological replicates (*n* = 5). Statistical analysis of proteome data from normal differentiating phBECs was performed at four timepoints (day 7, 14, 21, and 28) in comparison to day 0 using the Wald test with Storey correction to account for multiple testing to identify differentially expressed proteins (significance value q < 0.05).

**Figure 2 ijms-26-08324-f002:**
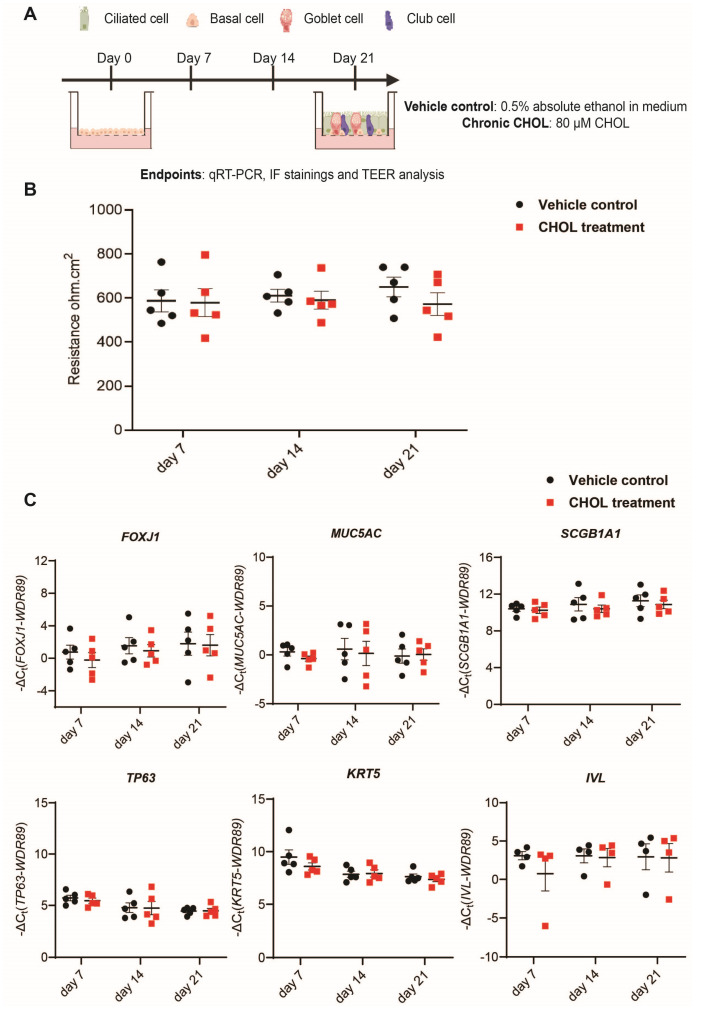
Chronic CHOL exposure does not affect epithelial barrier integrity and human bronchial epithelial cell type-specific transcripts during the differentiation phase. (**A**) Illustration of the chronic CHOL exposure via the basolateral compartment of the insert during the differentiation phase. (**B**) Epithelial barrier integrity was determined using TEER analysis after chronic CHOL exposure. Data presented as mean ± SEM (*n* = 5). (**C**) qRT-PCR analysis of transcript levels of human bronchial epithelial cell type-specific markers, such as the ciliated cell marker *FOXJ1*, the secretory cell markers *MUC5AC* and *SCGB1A1*, basal cell markers *TP63* and *KRT5*, and the squamous cell marker *IVL* after chronic CHOL exposure in vitro. WD Repeat Domain 89 (*WDR89*) was used as a housekeeping gene for samples obtained from CHOL-treated phBECs. Data presented as mean ± SEM (*n* = 5) except for the *IVL* transcript data, which was derived from *n* = 4. For Figure 2B,C, the statistical analysis was performed for chronic CHOL-treated phBECs in comparison to time-matched vehicle-treated phBECs using a two-tailed paired *t*-test with Bonferroni correction to account for multiple testing (no significant changes).

**Figure 3 ijms-26-08324-f003:**
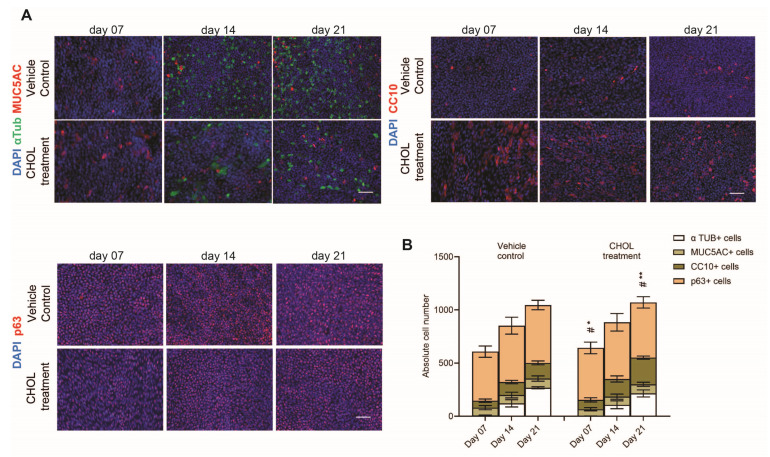
Chronic CHOL exposure leads to an increase in the CC10^+^ secretory cell population in vitro. (**A**) Representative IF stainings (scale bar: 50 µm) of human bronchial epithelial cell type-specific markers, such as alpha-tubulin (αTUB, ciliated cell marker), mucin 5AC (MUC5AC), club cell-specific protein 10 (CC10, encoded by *SCGB1A1*), and tumor protein 63 (p63), and (**B**) quantification of αTUB^+^, MUC5AC^+^, CC10^+^, and p63^+^ cells after chronic CHOL exposure for the indicated timepoints during the differentiation phase. Data presented as mean ± SEM (*n* = 5). The statistical analysis was performed for chronic CHOL-treated phBECs in comparison to time-matched vehicle-treated phBECs using a two-tailed paired *t*-test with Bonferroni correction to account for multiple testing. Significance is shown when *p*-value < 0.05 (*), and *p*-value < 0.01 (**). #: CC10^+^ secretory cells.

**Figure 4 ijms-26-08324-f004:**
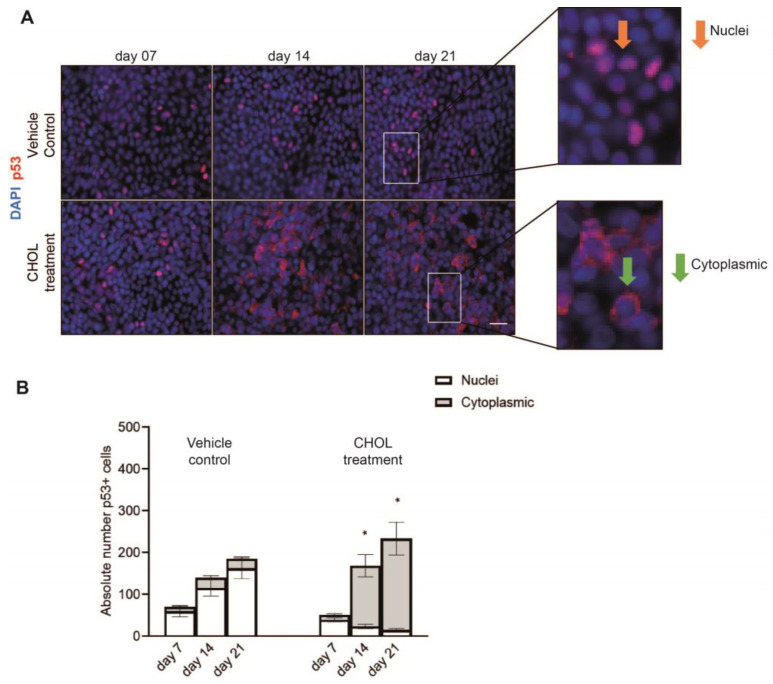
Chronic CHOL exposure affects the nuclear translocation of p53 in vitro. (**A**) Representative IF stainings (scale bar: 50 µm) and (**B**) related compartmental localization (nuclear or cytoplasmic) of tumor protein 53 (p53) after chronic CHOL exposure in comparison to time-matched vehicle control during the differentiation phase. Data presented as mean ± SEM (*n* = 5). The statistical analysis was performed for chronic CHOL-treated phBECs in comparison to time-matched vehicle-treated phBECs using a two-tailed paired *t*-test with Bonferroni correction to account for multiple testing. Significance shown when *p*-value < 0.05 (*).

**Table 1 ijms-26-08324-t001:** LC–MS/MS proteomic analysis parameters used to identify differentially expressed proteins involved in the CHOL biosynthesis pathway during normal phBECs differentiation. The MS/MS search parameters included a fragment mass tolerance of 20 mmu, a peptide mass tolerance of 10 ppm, and a false discovery rate (FDR) < 0.55%. The CHOL-metabolizing proteins were identified with at least two matching peptides, including one unique peptide, and Mascot confidence scores ≥ 47.

Accession	Peptide Count	Unique Peptides	Confidence Score	Gene Symbol	Description
Q9BWD1	13	10	407	ACAT2	Q9BWD1|THIC_HUMAN Acetyl-CoA acetyltransferase, cytosolic OS = Homo sapiens GN = ACAT2 PE = 1 SV = 2
P37268	12	10	409	FDFT1	P37268|FDFT_HUMAN Squalene synthase OS = Homo sapiens GN = FDFT1 PE = 1 SV = 1
Q15738	11	10	287	NSDHL	Q15738|NSDHL_HUMAN Sterol-4-alpha-carboxylate 3-dehydrogenase, decarboxylating OS = Homo sapiens GN = NSDHL PE = 1 SV = 2
Q16850	14	8	411	CYP51A1	Q16850|CP51A_HUMAN Lanosterol 14-alpha demethylase OS = Homo sapiens GN = CYP51A1 PE = 1 SV = 3
Q01581	10	8	341	HMGCS1	Q01581|HMCS1_HUMAN Hydroxymethylglutaryl-CoA synthase, cytoplasmic OS = Homo sapiens GN = HMGCS1 PE = 1 SV = 2
Q15392	8	7	268	DHCR24	Q15392|DHC24_HUMAN Delta (24)-sterol reductase OS = Homo sapiens GN = DHCR24 PE = 1 SV = 2
P14324	6	6	275	FDPS	P14324|FPPS_HUMAN Farnesyl pyrophosphate synthase OS = Homo sapiens GN = FDPS PE = 1 SV = 4
P53602	7	6	209	MVD	P53602|MVD1_HUMAN Diphosphomevalonate decarboxylase OS = Homo sapiens GN = MVD PE = 1 SV = 1
O00767	7	6	267	SCD	O00767|ACOD_HUMAN Acyl-CoA desaturase OS = Homo sapiens GN = SCD PE = 1 SV = 2
Q15800	4	4	141	MSMO1	Q15800|MSMO1_HUMAN Methylsterol monooxygenase 1 OS = Homo sapiens GN = MSMO1 PE = 1 SV = 1
Q9UBM7	3	3	68	DHCR7	Q9UBM7|DHCR7_HUMAN 7-dehydrocholesterol reductase OS = Homo sapiens GN = DHCR7 PE = 1 SV = 1
Q14534	2	2	54	SQLE	Q14534|ERG1_HUMAN Squalene monooxygenase OS = Homo sapiens GN = SQLE PE = 1 SV = 3
Q15125	2	1	89	EBP	Q15125|EBP_HUMAN 3-beta-hydroxysteroid-Delta (8), Delta (7)-isomerase OS = Homo sapiens GN = EBP PE = 1 SV = 3
Q14739	2	1	51	LBR	Q14739|LBR_HUMAN Lamin-B receptor OS = Homo sapiens GN = LBR PE = 1 SV = 2
O75845	2	1	47	SC5D	O75845|SC5D_HUMAN Lathosterol oxidase OS = Homo sapiens GN = SC5D PE = 1 SV = 2

**Table 2 ijms-26-08324-t002:** Upstream regulator analysis from normal differentiating phBECs. List of top upstream master regulators regulating the list of differentially expressed proteins from normal differentiating phBECs. The z-score estimates whether the upstream regulator activates or inhibits the differentially expressed target proteins in the dataset. A positive z-score suggests activation, while a negative z-score suggests inhibition. The *p*-value of overlap is a measure of statistical significance for the overlap between differentially expressed proteins from the dataset and the known targets of the mentioned upstream regulator.

Upstream Regulator	Molecule Type	Prediction Activation State	Activation Z-Score	*p*-Value of Overlap	Target Molecules in Dataset
TP53	transcription regulator	Activated	3.079	0.000000116	ACAA1, ACL, ACOX1, ACSL3, ADA, AK1, ALDH1A1, ALDH1A3, ARHGEF2, ARL6IP1, ARVCF, BIRC3, CCN2, CD276, CD44, CES2, CLU, COL18A1, COL1A1, COL3A1, COL7A1, CRYAB, CTSD, CYP51A1, DHCR7, DKK1, DUT, ECM1, EGFR, FAM§C, FASN, FDFT1, FDPS, FDXR, FEN1, FGFBP1, FOSL1, FTH1, GDF15, GLB1, HMGCS1, IFI30, IFI35, IFIT1, IGFBP7, IL1B, IL1RN, ITGB4, LAMC2, LBR, LGALS3, MAD2L1, MCM2, MCM3, MCM6, MCM7, ME1, MET, MPZL2, MTHFD2, MVD, NDRG1, NFKB2, NOTCH1, NUCKS1, OMA1, PFKP, PLAU, POLD1, PPP5C, PPT1, PRKCB, PRODH, PROM1, PUM3, RBM34, S100A9, SCD, SEL1L, SLC16A1, SLC2A1, SMC2, SMC4, SOD2, SPATA18, SQLE, STX6, TFRC, THBD, TIGAR, TLR3, TNFRSF10B, TUBB, VCAN
CST5	other	Activated	2.623	0.0000034	ACAT2, AHCTF1, ANK3, ANXA3, ARHGEF2, C17orf49, CD44, CNBP, CORO1A, CTSB, DDX21, DRAP1, EBNA1BP2, EEF1B2, EIF5B, FBX = 2, GALNT5, GDA, GPRC5A, INTS1, ITPRID2, LYAR, MSN, MYO5B, NIFK, NT5E, PITRM1, RBM28, RPF2, RRS1, RSL1D1, SSRP1, SURF6, TOMM40, VASP, VCAN, WDR3, WDR36
OGA	enzyme	Activated	2.558	0.0247	ACTG1, ALDOC, CDK5, CELSR1, CEP43, DHCR7, FDFT1, FDPS, FGFBP1, FLNA, FSCN1, GALK2, HMGCS1, LAMB1, MSMO1, NSDHL, PLAU, PROM1, S100A2, TNC, TP53BP1, TRADD
IFT88	other	Activated	2.449	0.000111	ACAT2, ACLY, CYP51A1, FDPS, MVD, NSDHL
KIF3A	enzyme	Activated	2.449	0.0000379	ACAT2, ACLY, CYP51A1, FDPS, MVD, NSDHL
ILF3	transcription regulator	Activated	2.132	0.0000013	ACAT2, ACTG1, CCDC170, CFH, DHCR7, EBP, EGFR, FADS2, FDPS, IL1B, IL1RN, LDLR, LGALS3, MCM5, NSDHL, PLAU, SCD, SLC3A2, STEAP4, TLR3, TNC
SREBF1	transcription regulator	Inhibited	−2.207	0.0000261	ACADS, ACLA, AK4, ALDOA, CAP2, CYP51A1, FASN, FDPS, GLB1, IFI30, LDLR, LGALS3, MDK, NPC1, OAT, P4HA2, PLEKHA4, PPAT, RNASET2, SCD, SERPINA3, TF
NUPR1	transcription regulator	Inhibited	−2.795	0.142	ACSS1, ALDH5A1, ALDOC, ANK3, BRI3BP, CCN1, CEBPB, COL3A1, DHCR24, ETV6, GDF15, GSTA4, HSPA2, LARS2, LBR, MKI67, NDRG1, NEK9, NUCKS1, NUP50, P4HA2, PARP9, PRR12, RAB32, RIBC2, RNF19B, SAMHD1, SHROOM3, SLC2A1, STK38, SUOX, TCTN2, TMPO, TNFAIP8L1, UPP1, ZNF512
THEM6	other	Inhibited	−3.302	0.000385	ACAA1, ASNS; CYP51A1, DHCR7, FDFT1, FDPS, HMGCS1, MSMO1, MVD, NSDHL, SQLE

**Table 3 ijms-26-08324-t003:** List of human primers used for qRT-PCR analysis.

Gene	Forward Primer Sequence (5′–3′)	Reverse Primer Sequence (5′–3′)
*FOXJ1*	TCGTATGCCACGCTCATCTG	CTTGTAGATGGCCGACAGGG
*IVL*	GGAGGTCCCATCAAAGCAAGA	GCTCCTTCTGCTGTGCTCA
*KRT5*	GAGATCGCCACTTACCGCAA	TGCTTGTGACAACAGAGATGT
*MUC5AC*	AGCAGGGTCCTCATGAAGGTGGAT	AATGAGGACCCCAGACTGGCTGAA
*SCGB1A1*	TTCAGCGTGTCATCGAAACCC	ACAGTGAGCTTTGGGCTATTTTT
*TP63*	CCCGTTTCGTCAGAACACAC	CATAAGTCTCACGGCCCCTC
*WDR89*	AGTACGTTCCATCCCAGCAATCC	AGGCCATCAGATGAACCTGAGACT

**Table 4 ijms-26-08324-t004:** List of antibodies used for immunofluorescence analysis.

Target	Host	Catalog Number	Provider	Dilution
αTUB	Rabbit	ab 179484	Abcam	1:500
CC10	Mouse	sc 365992	Santa Cruz	1:300
MUC5AC	Mouse	ab 3649	Abcam	1:250
p53	Mouse	sc126	Santa Cruz	1:100
p63	Mouse	ab 735	Abcam	1:125
Donkey anti-mouse (red 568)	Mouse	A10037	Thermofisher	1:500
Goat anti-rabbit (green 488)	Rabbit	A32731	Thermofisher	1:500

## Data Availability

All raw data supporting the findings of this study are available from the corresponding author, AC, upon reasonable request. The mass spectrometry proteomics data have been deposited to the ProteomeXchange Consortium via the PRIDE [1] partner repository with the dataset identifier PXD063763.

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
