# Peer review of "Cholesterol Regulates Airway Epithelial Cell Differentiation by Inhibiting p53 Nuclear Translocation"

_ijms, 2025, doi:10.3390/ijms26178324_

Round 1
Reviewer 1 Report
Comments and Suggestions for Authors
Comments are attached in the pdf.

Reviewer 2 Report
Comments and Suggestions for Authors
Introduction
1). First paragraph, seems rather poorly referenced. e.g. Line 49, no reference to support this statement. Line 56-59, no references to support the assertions made. Line 104/105 references missing. It is also very dense but lacks the specifics to understand your study design (e.g. what is the evidence that FOXJ1 is a specific marker for ciliated cells). Too many different concepts in one paragraph. How do club cells differ from squamous cells? The Introduction is not preparing the reader very well for why you have done the analyses you have.
2). Line 73, Cholesterol is not a neutral lipid.
3). Please change treatment to exposure.
4). Please also introduce the CC10+ secretory population. Also phBECs
5). Please also tell the reader (with supporting references to original research) how bronchial epithelial cells handle cholesterol uptake, and also whether they synthesize cholesterol.
Results
1). Figure 1 legend - What are ALI conditions.
2). Please provide in Supplementary Tables the list of all 4,860 detected proteins, and levels of all at each examined time point plus fold-changes and P values. Please also tell the reader which ones were altered at the early time point but at a later one, and vice versa. You have a lot of data but not really described it.
3). Please provide the results of the IPA in Supplementary tables. e.g. what genes in each category, fold changes. P values.
4). Results are too condensed. I would suggest providing a Figure of the cholesterol biosynthetic pathway and indicate which proteins had reduced values at which time points, and which did not.
5). Please provide references supporting your assertions made in this Introductory paragraph.
6). Line 151-154, I am not convinced that this is the way that bronchial epithelial cells would see cholesterol - would it not be via HDL- or LDL. This is why the Introduction requires improvement.
7). Please tell the reader more about the RT-qPCR assays. Were assays for each sample performed in duplicate or triplicate or not. Do biological replicates contain technical replicates. Western blots independently validating key proteins identified on the proteomic analysis would increase the robustness of the study results.
8). Section 2.2, the rationale for the analysis shown in Table 2 not provided. Nothing seems to quite link up. Why do the upstream analysis, and then not follow up the results with RNA or ChIP analyses? Close the circle, although I appreciate this will require additional experimental data. Which cells are responsive to cholesterol in terms of down-regulating key enzymes/receptors increasing cholesterol uptake, synthesis, storage and efflux.
Discussion, contains some Introduction.
Reviewer 3 Report
Comments and Suggestions for Authors
This study explores the role of cholesterol (CHOL) in regulating airway epithelial cell differentiation through inhibition of p53 nuclear translocation. The study addresses an important biological question and employs a multi-omics approach that includes proteomics, transcriptomics, and immunofluorescence. Overall, the experimental design is sound, and the findings contribute to our understanding of how lipid metabolism can influence epithelial cell fate.
Criticism/things that need to be improved/clarified:
- The authors should provide more context on the concentration of cholesterol used in the experiments and justify its relevance to physiological or pathological conditions.
- While the data suggest that cholesterol affects p53 nuclear translocation, the causal link remains indirect. I recommend either including additional mechanistic experiments (e.g., use of p53 modulators) or discussing this limitation more explicitly.
- Given that cholesterol is a major structural component of the plasma membrane, it would be valuable for the authors to briefly discuss how changes in membrane composition might contribute to the observed effects on p53 localization and epithelial differentiation.
- Cholesterol’s role in organizing membrane microdomains (e.g., lipid rafts) could influence receptor-mediated signaling or nuclear import mechanisms. A short discussion on this aspect would enhance the mechanistic depth of the manuscript and link the molecular findings more clearly to cholesterol’s established membrane functions.
